# Exosomal microRNA in Pancreatic Cancer Diagnosis, Prognosis, and Treatment: From Bench to Bedside

**DOI:** 10.3390/cancers13112777

**Published:** 2021-06-03

**Authors:** Md. Hafiz Uddin, Mohammed Najeeb Al-Hallak, Philip A. Philip, Ramzi M. Mohammad, Nerissa Viola, Kay-Uwe Wagner, Asfar S. Azmi

**Affiliations:** Departments of Oncology, Karmanos Cancer Institute, Wayne State University School of Medicine, Detroit, MI 48201, USA; uddinh@wayne.edu (M.H.U.); alhallakm@karmanos.org (M.N.A.-H.); philipp@karmanos.org (P.A.P.); mohammad@karmanos.org (R.M.M.); violan@karmanos.org (N.V.); wagnerk@karmanos.org (K.-U.W.)

**Keywords:** pancreatic cancer, biomarker, exosome, microRNA (miRNA), early diagnosis, prognosis, treatment

## Abstract

**Simple Summary:**

Pancreatic cancer is the fourth leading cause of cancer death in the United States and over 90% of the patients suffer from pancreatic ductal adenocarcinoma (PDAC). PDAC is the most lethal gastrointestinal malignancies and only 10% of the people survive more than 5 years, therefore, novel diagnostic, prognostic, and therapeutic strategies are an immediate necessity. Studies have demonstrated microRNAs in bodily fluids that are bound with membranes (exosomes) can act as stable biomarkers both for disease development and metastasis. The diagnostic, prognostic, as well as therapeutic roles of exosomal microRNAs in pancreatic cancer have been discussed in this review.

**Abstract:**

Pancreatic cancer is the fourth leading cause of cancer death among men and women in the United States, and pancreatic ductal adenocarcinoma (PDAC) accounts for more than 90% of pancreatic cancer cases. PDAC is one of the most lethal gastrointestinal malignancies with an overall five-year survival rate of ~10%. Developing effective therapeutic strategies against pancreatic cancer is a great challenge. Novel diagnostic, prognostic, and therapeutic strategies are an immediate necessity to increase the survival of pancreatic cancer patients. So far, studies have demonstrated microRNAs (miRNAs) as sensitive biomarkers because of their significant correlation with disease development and metastasis. The miRNAs have been shown to be more stable inside membrane-bound vesicles in the extracellular environment called exosomes. Varieties of miRNAs are released into the body fluids via exosomes depending on the normal physiological or pathological conditions of the body. In this review, we discuss the recent findings on the diagnostic, prognostic, and therapeutic roles of exosomal miRNAs in pancreatic cancer.

## 1. Introduction

The pancreas is a complicated dual-functional organ that links the digestive system with the endocrine system [1]. Due to the variations in tissue types within the pancreas, it gives rise to tumors of multiple origins broadly categorized into exocrine and endocrine tumors [2]. Among them, pancreatic ductal adenocarcinoma (PDAC) is responsible for the highest morbidity not only in exocrine tissue-derived malignancy but also among all pancreatic cancer types [3]. PDAC accounts for more than 90% of all pancreatic malignancies [4,5]. According to the National Cancer Institute, in 2020 there were 56,770 new cases of PDAC and 45,750 deaths, which makes PDAC as the fourth leading cause of cancer-associated deaths in the United States affecting both men and women [3]. According to the American Cancer Society, the five-year overall survival rate of PDAC patients between 2010 and 2016 was about 10% [6]. PDAC can be localized, locally advanced invasive, or metastasized to distant organs, [3] most commonly the liver [7,8]. Pancreatic cancer patients are frequently characterized with noteworthy mutations in four essential genes, including *KRAS*. In PDAC patients, the mutational activation of the proto-oncogene *KRAS* occurs in 85–90% of cases [9,10,11]. The hotspot residue for the KRAS mutation is G12. The prevalence of *KRAS*^G12D^, *KRAS*^G12V^, *KRAS*^G12R^, and *KRAS*^G12C^ mutations in PDAC are 39.2%, 32.5%, 17%, and 1.7%, respectively [12]. Though *KRAS*^G12C^ mutations are only observed 1.7% of PDAC patients, its protein product has been extensively studied over the past few years and targeted with covalent inhibitors such as AMG510, MRTX849 [10]. Alterations in other tumor suppressor genes include *CDKN2A* [13], *TP53* [14], and *DPC4/SMAD4* [15]. The rate of mutations in *CDKN2A*, *TP53*, and *DPC4/SMAD4* is over 95%, up to 75%, and about 55%, respectively [16].

Surgical removal of the tumor mass is considered as the only curative treatment of PDAC [17,18], but most cases are being diagnosed at more advanced stages [17,19] and only 15–20% of patients are eligible for surgery [17,18]. Despite the success of surgery, a large number of patients experience disease recurrence either locally or distantly within a year, keeping the five-year survival rate as low as 20% [20]. A number of pathological factors, such as resection margin status as well as lymph node and perineural invasions, are associated with such recurrence [21,22,23,24]. The invasion of PDAC into the peripancreatic tissues, including adipose, also has been recognized as a cause of such failure [25,26,27,28]. Unfortunately, no early diagnostic biomarker or imaging modality for PDAC is currently available. The carbohydrate antigen 19-9 (CA 19-9) is frequently elevated in 75–85% of PDAC patients [29], but it lacks sensitivity (80%) and specificity (73%) as a biomarker of PDAC [17,19,30]. Furthermore, 5–10% of the population cannot synthesize CA 19-9 due the absence of enzyme 1,4-fucosyl transferase, which is required for antigen epitope production [31]. Tragically, treatment options for most PDAC patients with metastasis are very limited [32], therefore, the development of diagnostic and prognostic biomarkers, along with a new therapeutic strategy for PDAC, is an urgent necessity.

Recent findings demonstrated a critical role of the tumor microenvironment in the development and progression of PDAC [33,34,35]. One of the major players in tumor microenvironment-associated intercellular communication is exosomes. The cargo of exosomes contains lipids, proteins, metabolites, and genetic material, including DNA, mRNA, microRNAs (miRNAs), and long non-coding RNAs (lncRNAs) [36,37,38]. Mounting evidence has revealed that exosomal miRNAs are intensely connected with various cancers, including pancreatic cancer [39]. In the past few years, between 2014 and 2018, an effort has been made to determine the suitability of exosomal miRNAs as diagnostic/prognostic biomarkers and their potential use in the treatment of PDAC [40,41,42]. In this current review, we discuss recent advancements in diagnosis, prognosis, and therapeutics for pancreatic cancer utilizing novel findings of exosomal miRNAs.

## 2. Exosomes

### 2.1. Structural Composition of Exosome

Exosomes are membrane-bound extracellular vesicles (EVs) containing biological materials and play an important role in communication among cells [43]. The ultrastructure and origin of exosomes were explained in the 1980s by several groups [44,45,46]. They are formed by the invagination of multivesicular bodies (MVBs) and subsequent intraluminal vesicles (ILVs) formation via inward budding [44,47]. Exosomes are released in the extracellular space through a process of fusion with the plasma membrane [44]. The size of exosomes varies from 30 to 150 nm in diameter with an average of 100 nm and is restricted by the size of MVBs [37,48,49,50]. Almost all types of cells in the human body secrete exosomes in diverse body fluids, including blood, urine, saliva, bile, ascites, breast milk, cerebrospinal fluid, as well as within the tissue matrix [48,49,51,52,53,54]. The secretory exosomal cargo varies depending on the cell types as well as the local or systemic physiology or pathology [51,52], such as reproduction, immune response, fibrosis of tissue, oncogenesis, and the metastatic spreading of cancer cells [55,56,57,58].

The term “exosomes” was proposed by Johnstone et al. in 1987 [59]. Earlier studies showed that exosomes that are being released from B lymphocytes transformed with the Epstein–Barr virus can trigger T-cell responses [60]. Exosomes can be discharged from almost all prokaryotic and eukaryotic cell types [37,43,61,62,63]. They also can be identified in the conditioned medium of cell cultures [50,64,65]. Exosomes are comprised of various proteins, including tetraspanin (CD9, CD81) [66], heat-shock proteins (Hsp70, Hsp90) [67], members of the endosomal sorting complexes required for transport (ESCRT), such as Alix and Tsg101, cytoskeletal proteins (actin, tubulin), as well as GTPases [66]. These proteins take part in the biogenesis, sorting, and release of exosomes [68]. They also play an important role in the presentation of antigens, the modulation of the cytoskeleton, endosomes, and microdomain membranes [69].

### 2.2. Isolation and Identification of Exosomes

All exosomes are extracellular vesicles, but not all extracellular vesicles are exosomes. Once outside the cell membrane, their origin is difficult to determine [61,70]. Therefore, the International Society for Extracellular Vesicles (ISEV) has specified a few basic prerequisites for the isolation and identification of exosomes [61]. So far, several surface makers have been established, most belonging to the category of tetraspanin, including CD63, CD81, and CD9. Other markers involve Alix, Flotillin-1, Syntenin-1, and TSG101 [43]. Though the ISEV has not determined any gold standard for the isolation of exosomes, they recommended the inclusion of detailed methodologies in scientific publications to ensure reliability and reproducibility. This may be due to the fact that an isolation of pure exosomes is difficult due to the presence of soluble proteins as contaminants in any body fluids [71,72]. The most widely used methods are ultracentrifugation, size exclusion chromatography, polymer precipitation, and immunocapture [73,74]. In the years 2018 and 2020, more efficient methods had been developed, such as the asymmetric flow field-flow fractionation (AF4) technique and integrated microfluidic chip (IMC) method, which are able to isolate high-quality exosomes [75,76]. The AF4 protocol can separate exosome subsets, such as large vesicles (90 nm–120 nm), small vesicles (60 nm–80 nm), as well as nanoparticles of 35 nm size. On the other hand, the IMC platform allows isolation of exosome subsets by surface markers, such as CD63. Besides these two methods, an acoustic nano filter system also has been developed that can separate vesicles in a continuous and contact-free way [77].

### 2.3. Role of Exosomes

The communication between cells in the tumor microenvironment carried out by exosomal miRNAs is performed via complicated signaling events [37,78,79,80]. Exosomes mediate the communication to a particular cell type, primarily via receptor–ligand interaction or receptor-mediated endocytosis, such as clathrin-coated pits, lipid rafts, phagocytosis, or micropinocytosis [81]. Many investigations are ongoing to decipher the role of exosomes in cell-to-cell signaling or intercellular communication [82]. The transfer of mRNAs and miRNAs through exosomes occurs via receptor-mediated endocytosis [81] and has been reported in an earlier study on various types of cells [83], pointing to a novel regulatory mechanism in exosome-mediated cell-to-cell communication. In fact, adaptive immune response to tumors by the exosomal cargo of certain immune cells (e.g., dendritic cells) has been demonstrated recently in 2016 [84]. Such findings prompted an ample number of studies in the avenue of exosomal regulation over the past 10 years [85,86,87,88,89,90,91]. Exosomes facilitate communication among cells via autocrine, paracrine, or even by endocrine signaling mechanisms [52]. Exosomes released by cancer cells can travel to distant organs, such as the liver and brain, and can modulate the microenvironment to establish a metastatic niche and subsequent metastasis [7,8].

In 2005, exosomal regulation of migration and cytotoxicity of natural killer (NK) cells was shown in a Colo357 PDAC cell line model for the first time [92]. Subsequently, various studies have revealed the functional role of exosomes in pancreatic cancer, particularly in proliferation, metastasis, pathobiology, and drug resistance [8,92,93,94]. Masamune et al. has demonstrated the exosomal regulation of profibrogenic activites in pancreatic stellate cells likely due to the overexpression of miR-1290 [95]. Pancreatic cancer cell-derived exosomes have been shown to be associated with the onset of diabetes mellitus [96,97,98] and lipolysis-mediated weight loss [99]. In 2015, a published report showed that the Kupffer cells of the liver are highly influenced by the macrophage migration inhibitory factor positive (MIF+) exosomes of PDAC cells in mice. The PDAC-derived exosomes activate Kupffer cells via MIF, which in turn induces the release of transforming growth factor β (TGF-β). The TGF-β secretion subsequently promotes fibronectin release by hepatic stellate cells (HSC) and migration of PDAC cells into the liver. The PDAC cell-derived exosomes overexpress MIF and its knockdown reverse liver metastasis [8]. It has been demonstrated that KRAS mutation was present in the histologically negative surgical margins of 37 of 70 (53%) patients with pancreatic cancer [100]. Such delivery of KRAS mutation might occur via exosomes. Though exosome contains proteins, lipids, nucleic acids, most of the studies repeatedly demonstrated the role of exosomal miRNAs in progression and drug resistance in pancreatic and other human cancers [101,102,103,104].

## 3. Exosomal miRNAs and Pancreatic Cancer

The miRNAs are small, noncoding RNAs of ~19–24 nucleotide length that act on mRNA for silencing or degradation and post-transcriptional regulation. They bind completely or partially to 3′-UTR, the untranslated regions of mRNA, via the base-pairing principle [83,105,106,107,108]. In humans, miRNAs regulate ~70% of the mRNA transcripts. Hence, it can be assumed that the miRNAs are involved in most of the cellular processes, including the development of tumors [108,109]. The changes in miRNA expressions are frequently reported in many human cancers, including PDAC [105,110,111,112]. Differential expression of miRNAs is observed in normal pancreatic ductal cells and pancreatic cancer cells. So far, many studies have investigated circulating miRNAs for their suitability as biomarkers [113,114,115,116,117,118]. The elevation of about twenty circulatory miRNAs has been demonstrated in pancreatic cancer patients. The miRNA panel detected in plasma served as promising biomarkers for the diagnosis of pancreatic cancer [119,120,121,122,123,124,125,126]. For being one of the most important cargos of exosomes [83], exosomal miRNAs went through excessive profiling in different cancers, including pancreatic cancer [127,128,129,130,131,132].

Highly elevated levels of exosomal miR-17-5p and miR-21 were detected in serum samples of pancreatic cancer patients for the first time in 2013 [133]. Since then, several studies have explored the association of exosomal miRNAs in different stages of pancreatic cancer development [38]. In pancreatic cancer cells, Sun et al. showed that IL-26 level was regulated by the exosomes loaded with miR-3607-3p secreted by natural killer (NK) cells, which suppressed disease progression in vivo [134]. It has been demonstrated that activated pancreatic stellate cells (PSCs) continuously release miR-21-enriched exosomes that are endocytosed by pancreatic cancer cells. The abundant miR-21 inside cancer cells activates RAS/ERK signaling and induces epithelial-to-mesenchymal transition (EMT) [135]. Pancreatic cancer cells are able to halt their cell cycle at the G1/S phase by absorbing exosomal miR-194-5p from the dying cells resulting from radiotherapy. This allows cancer cells to repair their DNA and repopulate with residual tumor cells [136]. The exosomal miRNA-mediated regulations of pancreatic cancer are summarized in Figure 1.

## 4. Exosomal miRNAs as Diagnostic Biomarkers for PDAC

There are many promising circulating miRNA biomarkers that have been proposed over the last few years for pancreatic cancer. However, due to the heterogeneous nature of the circulating miRNAs, it has not yet been possible to establish any miRNA-based detection method for the clinic either for early diagnosis or for the detection of pancreatic cancer [37,137]. The forms of miRNAs in the circulations, such as free, protein-bound, or those associated with exosomes, as well as their origin cell types mediate heterogeneity. Such diverse forms add more variability in sensitive and specific detection of miRNAs in the biological fluids of pancreatic cancer patients. Furthermore, the stability, selective isolation, and detection of miRNAs in circulation associated with pancreatic cancer make the establishment of miRNAs as biomarkers more difficult [3]. The various limitations of circulating miRNAs can be overcome by analyzing circulating exosomal miRNAs, which are intrinsically protected against various degradation processes and expected to be enriched in cancer due to the shedding of exosomes from primary tumor sites, as evident from different in vitro and in vivo systems [138,139]. There are several exosomal miRNAs found in pancreatic cancer that could serve as potential biomarkers for the diagnosis or early tumor detection (Table 1) and described below.

### 4.1. Exosomal miRNA in Blood Samples

Blood serum is widely used for the isolation of exosomes from cancer patients. As mentioned above, serum exosomal miR-17-5p and miR-21 were the first indication of a differential overexpression of miRNAs in pancreatic cancer as compared to non-malignant pancreatic diseases as well as healthy controls [133]. Sometime later, a panel of serum exosomal miRNAs involving miR-1246, miR-4644, miR-3976, and miR-4306 was proposed as a potential diagnostic biomarker for pancreatic cancer [140]. A combination of upregulated and downregulated miRNA panels was predicted to act as a biomarker for the detection of pancreatic cancer. The panel includes miR-10b, miR-21, miR-30c, and miR-181a that are overexpressed and miR-let7a that is underexpressed in pancreatic cancer. The sensitivity and specificity of this panel (100% and 100%, respectively) were better than exosomal glypican-1 (~34% and 100%, respectively) or conventional CA 19-9 (86% and 100%, respectively) [141]. The study observed elevated exosomal miR-10b, miR-21, and miR-30c in all 29 PDAC patients whereas only in 8 cases showed a normal level or slight CA 19-9 increase [141]. Exosomal miR-191, miR-21, and miR-451a also showed significant elevation in pancreatic cancer, including the intraductal papillary mucinous neoplasm (IPMN) subtype [142]. Additionally, differentiation between pancreatic cancer and IPMN patients could be possible by utilizing the expression of plasma miR-483-3p [143].

Plasma exosomal miRNAs serve as non-invasive biomarkers for various types of cancer [144]. Like in serum, exosomal miR-21 was also detected from plasma samples and was able to distinguish PDAC and intraductal papillary mucinous neoplasm when compared to a healthy control [142,149]. A couple of plasma exosomal miRNA biomarkers, namely miR-196a and miR-1246, have been demonstrated to be elevated in non-metastatic pancreatic cancer and have the potential for early detection [130,131]. For the identification of stage I pancreatic cancer, a combination of exosomal miRNAs (miRNA-16a and miRNA-196a) and traditional CA-199 showed a strong statistical association [145,146], which indicates the importance of miRNAs as peripheral biomarkers for pancreatic cancer [162].

### 4.2. Exosomal miRNA in Other Body Fluids

To identify novel diagnostic biomarkers for pancreatic cancer, the most relevant biological fluid, the pancreatic juice, was utilized in a study for the isolation of exosomal miRNAs [147]. The study detected miR-21 and miR-155 in the isolated exosomes from the pancreatic juice and showed their utility as potential biomarkers for this malignancy [147]. The pancreatic-juice-derived exosomal miR-21 levels were significantly different between PDAC and chronic pancreatitis patients [147]. Exosomal miRNAs also have been isolated from saliva to find novel biomarkers in various cancers [148,163]. Machida et al. have reported salivary exosomal miR-1246 and miR-4644, which are linked with pancreatobiliary tract cancer [148].

A clinical trial (NCT04636788) was initiated at the end of the year 2020 to identify potential exosomal miRNA biomarkers for early detection of pancreatic cancer among other small RNAs in combination with EUS-FNA tissues. The trial is still recruiting patients and healthy controls with a target of 102 participants. A total of 12 mL venous blood will be collected from participants and the study will assess miRNAs and other small RNAs by next-generation sequencing. Primary outcome measures include the sensitivity and specificity, and secondary outcome measures include the survival of patients.

Though several exosomal miRNA biomarkers showed promise for the diagnosis of pancreatic cancer, no universal exosomal miRNA signatures have been recognized so far. This is due to the lack of reproducibility of the results obtained in different studies and associated with exosome isolation methods, utilized models, and miRNA analysis tools [37]. Therefore, necessary strategies are needed to reduce the variability among model systems, exosome isolation techniques from different sample types, and further processing of samples. Taken together, a single exosomal miRNA can provide a higher specificity while being less sensitive. A combination of exosomal miRNAs can overcome the limitation of the current traditional early detection approach for pancreatic cancer [164]. Furthermore, exosomal miRNAs, along with proteins and conventional blood biomarkers, will improve the sensitivity and specificity of diagnosis undoubtedly.

## 5. Exosomal miRNAs as a Prognostic Biomarker in PDAC

The exosomal miRNA-mediated cell-to-cell signaling in the tumor microenvironment plays a significant role in the progression of cancer [165,166]. In the recipient cells, exosomal miRNAs cause a differential regulation of target genes, inducing angiogenesis, immune response, and metastasis [78,79,167]. In the primary tumor exosomes, selective miRNAs are enriched, exceeding the levels of normal cells that affect specific signaling pathways associated with cancer progression [78,129,131,168]. Specifically, the miR-451a in exosomes isolated from pancreatic cancer patient plasma samples was shown to be linked to the prognosis of this malignancy. The study analyzed the data using a miRNA microarray technique and was able to predict disease recurrence [144,149]. In a mechanistic study of pancreatic cancer, it has been demonstrated that enriched miR-21 and miR-221 take part in the crosstalk among pancreatic cancer cells, pancreatic stellate cells, and cancer-associated fibroblasts [169]. It is speculated that such crosstalk was governed by the release of exosomal miRNAs from cancer cells to neighboring cells.

In an in vitro study, an elevated level of miR-23b-3p was detected in the exosome derived from pancreatic cancer cells. Further investigation with the overexpression of miR-23b-3p confirmed the involvement of this miRNA in cell growth, migration, and invasion [150]. A similar study in pancreatic cancer cells found exosomal miR-339-5p in the regulation of migration and invasion activities [151]. The proliferation and invasive properties of surrounding cancer cells were shown to be modulated by pancreatic cancer exosomal miR-222 [52]. In this study, Li et al. reported that miR-222 regulates and relocates p27 to induce the proliferation and invasion of cancer cells [52].

Pancreatic cancer cells were shown to release exosomes enriched with miR-301a under hypoxic conditions. Mechanistically, miR-301a was found to contribute to the polarization of M2 macrophages via PTEN/PI3K signaling and the metastasis of pancreatic cancer cells [104] (Figure 1). In the tumor microenvironment of pancreatic cancer, M2 macrophages can release exosomes enriched in miR-501-3p. MiR-501-3p suppresses TGFBR3, a tumor suppressor gene, and promotes pancreatic cancer development and progression via the activation of the TGF-β signaling pathway [155] as well as IL10 and arginase secretion [104].

In contrast to a cancer-promoting role of exosomal miRNAs, a recent pancreatic cancer study found that exosomes derived from bone marrow mesenchymal stem cell (BMSC) contain elevated levels of miR-126-3p, which subsequently downregulate ADAM9 and suppress cancer progression [157]. The BMSC-derived exosomal miRNA-1231 was also shown to suppress the development of pancreatic cancer [158]. Moreover, exosomal miR-145-5p derived from human umbilical cord mesenchymal stromal cells was reported to reduce the progression of pancreatic cancer [159]. A study also demonstrated that potential miRNA biomarkers derived from portal vein blood exosomes (miR-4525, miR-451a, and miR-21) can be utilized for the evaluation of pancreatic cancer recurrence and overall survival [149]. Angiogenesis is one of the major cancer hallmarks. It is increasingly clear that exosomes have a role in the regulation of angiogenesis in pancreatic cancer [170]. In vitro and in vivo studies conducted by Shang et al. demonstrated that exosomal-derived miR-27a from pancreatic cancer cells promotes angiogenesis of human microvascular endothelial cells via BTG2 [152].

The abundance of certain exosomal miRNAs can be utilized to assess therapy response. For example, plasma miR-221 levels are elevated in pancreatic cancer patients after just three weeks of lapatinib and capecitabine treatment. Further evaluation of data showed a relation to drug resistance [171]. Exosomes from lapatinib- and capecitabine-treated pancreatic cancer patients likely contain elevated levels of miR-221. Therefore, exosomal miR-221 may serve as a potential biomarker to predict therapy response against these two drugs during the treatment period. The most potential prognostic biomarkers are summarized in Table 1.

In addition to the cancer suppressive and progressive roles, generalized immunosuppression also can be accomplished by exosomal miRNAs in pancreatic cancer via the regulation of MHC-II molecule expression. Pancreatic cancer cells can transfer miR-212-3p to dendritic cells by releasing miR-212-3p-enriched exosomes. Consequently, it can suppress the expression of regulatory factor X-associated protein, leading to a downregulation of MHC-II molecules [153,154].

Recent studies have constructed the foundation of exosomal miRNA biomarker research for the assessment of proliferation, migration, and invasion in pancreatic cancer. However, the exchange mechanism of exosomes between cancer cells and stromal cells and how they regulate recipient cells remains to be defined. Sufficient longitudinal studies are essential for the proper identification of exosomal miRNA as prognostic biomarkers in PDAC.

## 6. Exosomal miRNAs in the Treatment of PDAC

### 6.1. Exosomes as a Drug Delivery System

In 2020, engineered exosomes loaded with a cargo of choice have been utilized as a drug delivery system [172,173]. The drug-carrier exosomes have several advantages over other drug delivery approaches, such as liposome- and nanoparticle-mediated delivery. The carrier exosomes are biocompatible, well-tolerated, and induce less toxicity or immune reaction when isolated from and delivered to patients [174,175,176]. A recent in vivo study has demonstrated the ability of exosomes to cross the blood–brain barrier and deliver siRNAs into the brain [177]. Evidence also showed the absorption of therapeutic exosomal content by target cells in a mouse model when introduced intravenously. The observed immune clearance rate in the study was low, further suggesting its biocompatibility [175,178]. Additionally, toxic side effects have not been observed in vivo even after repeated injection of exosomes derived from mesenchymal or epithelial cells [42]. The tolerability of exosomes collected from mesenchymal stem cells was demonstrated by Kordelas et al. [179] when used for the treatment of graft-versus-host disease.

### 6.2. Exosomal miRNA Therapy in Pancreatic Cancer Progression and Therapy Resistance

A growing body of research findings suggests that exosomes can be used for treating pancreatic cancer in the clinic for disease stabilization or reduction of aggressiveness [180]. The use of exosomes has been drawing intense attention because of their capability to transfer diverse biomolecules, including miRNAs, that may modulate the tumor microenvironment in an interactive fashion [180].

Several studies convincingly suggest that exosomal miRNAs are linked to pancreatic cancer therapy resistance. For example, standard therapeutic choice gemcitabine is usually administered for the treatment of pancreatic cancer. However, upon treatment it causes cancer-associated fibroblasts (CAF) to release exosomal miR-146a and miR-106b to modulate the transcription process [160,161]. Mechanistically, exosomal miR-106b was shown to target TP53INP1 to activate downstream signaling [161]. These studies support the notion that CAF-derived exosomes can induce gemcitabine resistance in pancreatic cancer cells [160] (Figure 1). Both in vitro and in vivo studies on pancreatic cancer cells found a strong association of higher expression of exosomes miR-155 and gemcitabine resistance [101]. One of the mechanisms of this resistance could be the suppression of DCK, which is a key gemcitabine-metabolizing gene, via exosomal miR-155 transfer to recipient cancer cells. The silencing and overexpression study of miR-155 and DCK, respectively, proved their involvement in resistance [102].

In a study, gemcitabine-resistant pancreatic cancer stem cells were developed to study exosomal miRNA’s role in resistance. The study provided evidence that exosomal miR-210 can induce horizontal transfer of drug-resistant abilities against gemcitabine from resistant to sensitive cells [156]. Though current knowledge on the role of exosomal miRNAs in gemcitabine resistance is limited, these studies pointed out several new molecular targets to overcome gemcitabine resistance in pancreatic cancer through exosomal miRNA modulation.

### 6.3. Possibility of Exosomal miRNA to Target Key Mutations in Pancreatic Cancer

The initiation of pancreatic cancer is a multi-step process that starts with the mutations of certain genes, such as *KRAS*, *TP53*, and *CDKN2A* [181]. More than 90% of cases show activating *KRAS* mutations in pancreatic cancer [182] that are being considered attractive targets by any means, including through exosomal regulation. In 2017, engineered exosomes loaded with siRNAs were first introduced in a mouse model for PDAC. Clinical-grade exosomes were isolated from *KRAS*-G12D mutated mesenchymal stem cells and loaded with a specific siRNA that targets *KRAS*-G12D. This proof-of-principle study provided evidence that the expression of mutant *KRAS*-G12D can be downregulated through exosome-mediated knockdown of the mutant gene in a mouse model for PDAC [41]. The animals survived for a significantly longer time (increased overall survival rate) with no toxic adverse events [41]. These encouraging results led to a Phase I clinical trial in PDAC patients with a *KRAS^G12D^* mutation (NCT03608631) [38]. After subsequent verification of their safety and efficacy in organoid and patient-derived tumor xenograft models, it can be expected that engineered exosomes can be used as vehicles to deliver miRNAs as therapeutic agents for the targeted treatment of PDAC.

Thus far, the use of engineered exosomes in pancreatic cancer is still in its infancy. In the year 2014, investigations tested intravenous delivery of miRNA or siRNA-loaded exosomes to inhibit the aberrant signaling of PDAC cells. The study suggests transportation of miRNA or siRNA to the target cells would occur without degradation as they are protected inside lipid bilayered exosomes from natural ribonuclease present in the blood [183]. There are ample studies that identified a number of miRNAs for targeting pancreatic cancer [184,185]. They have been extensively reviewed elsewhere [186,187,188,189]. Xia et al. demonstrated that miR-7 can block pancreatic cancer via suppression of cell proliferation, migration, and invasion. In a pre-clinical model, a study observed an inhibition of tumor growth when miR-7 is overexpressed. At the molecular level, miR-7 targets MAP3K9 directly to exert its suppressive role [190]. On the other hand, miR-182 has been shown to target the cell cycle checkpoint regulatory molecule β-TrCP2 to promote cell proliferation and migration. This finding suggests a significant role of miR-182 in pancreatic cancer progression [191]. Additionally, miRNAs such as miR-7 [190], miR-205 [184], or miR-182 [191] can be included in the cargo of engineered exosomes and checked for the treatment efficacy and toxicity of preclinical model systems. Using such a strategy, it can be expected that the delivery of exosomal miR-7 will have more pronounced effect in the treatment of pancreatic cancer.

## 7. Conclusions and Future Perspective

Pancreatic cancer is a highly aggressive and lethal malignancy mostly due to its late-stage presentation. This malignancy is difficult to diagnose, monitor, and treat, hence the development of novel diagnostic and prognostic biomarkers and better therapeutic strategies are urgently needed. Several groundbreaking discoveries over the past decade on cancer-associated exosomes demonstrated an association between exosomal miRNA and the development, progression, and therapy-resistance of pancreatic cancer. Despite the reproducibility problem, several exosomal miRNAs, alone or in combination, showed superior potential as non-invasive diagnostic and prognostic biomarkers for pancreatic cancer. The International Society for Extracellular Vesicles has not declared precise standards for the isolation and processing of exosomes yet [192], but for the uniformity of findings in different studies, setting up a gold standard isolation method is crucial.

The area of exosome biogenesis and the mechanisms of exosome sorting and release in pancreatic cancer and other malignancies need to be explored further to gain a better understanding of the cellular processes that drive cancer progression and metastases that are being mediated by miRNAs and other cargos (e.g., growth factors). Members of the endosomal sorting complex required for transport 1 (ESCRT-1) are crucial for the intracellular routing of a subset of cargo proteins that are destined for inclusion into exosomes. One of those ESCRT-1 factors is the tumor susceptibility gene 101 (TSG101), which was identified within extracellular vesicles almost 20 years ago and is now being employed as one of the markers for exosome isolation [61,193]. While more recent studies have shown that targeting TSG101 can inhibit the release of exosomes [194], the biological significance of this and other ESCRT proteins in exosome- and miRNA-mediated processes that promote inflammation and cancer has yet to be explored [195].

In future studies, the molecular and cellular processes by which miRNAs as exosomal cargoes exert their effects on the growth and invasive properties of cancer cells should also be examined under hypoxic experimental conditions. As discussed earlier, some cancer-relevant miRNAs are exclusively upregulated in hypoxic cancer cells. While most studies are being performed under normoxic conditions, the experimental results and may not be readily applicable to in vivo cancer models or humans. Another under-studied area is the feedback loop of exosomal-miRNA-based regulation besides unidirectional signaling, which also requires further attention. So far exosome-based treatment is limited to in vitro and in vivo experiments for pancreatic cancer. Besides preclinical, clinical investigations are essential to identify universal exosomal miRNA signatures for the diagnosis, prognosis, and treatment of pancreatic cancer. In summary, extensive research on exosomal miRNAs in pancreatic cancer is needed to overcome current obstacles and guide current progress ahead toward a novel biomarker discovery and a new therapeutic avenue.

## Figures and Tables

**Figure 1 cancers-13-02777-f001:**
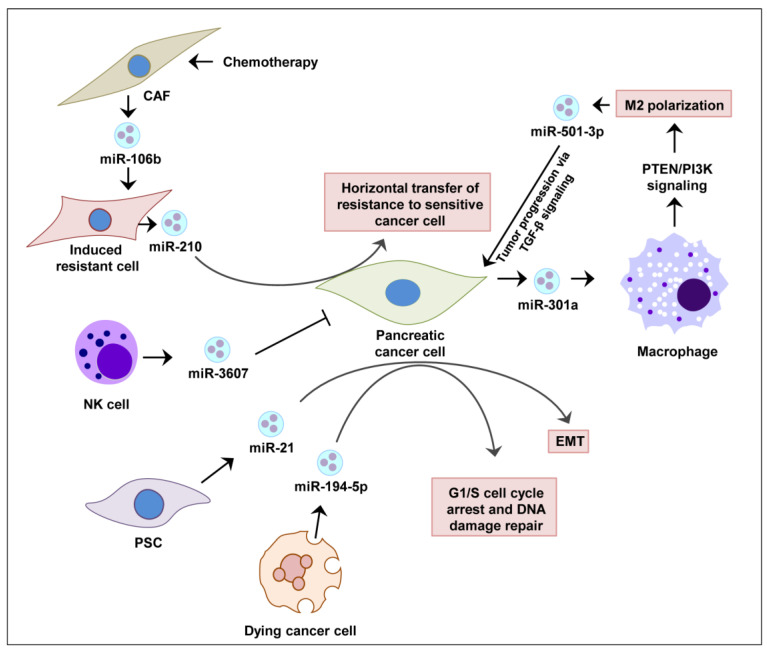
Role of exosomal miRNAs in the pancreatic cancer tumor microenvironment. In response to drug (gemcitabine chemotherapy), CAF releases exosomes enriched in miR-106a, which are absorbed by the pancreatic cancer cells. Expression of miR-106a-induced therapy resistance against gemcitabine. The resistance cell can transfer is resistant properties to gemcitabine-sensitive cells via the transfer exosomal miR-210 horizontally. NK cells can inhibit the pancreatic cancer cell’s proliferation through the secretion of exosomal miR-3607. PSC cells release exosomes abundant in miR-21 in the pancreatic tumor microenvironment, which promotes the expression of EMT-associated genes and subsequently promotes metastasis. Also dying cells in the pancreatic tumor microenvironment release miR-194-5p-containing exosomes, which results in G1/S phase cell cycle arrest of pancreatic cancer cells. The arrested cells spend sufficient time to repair their DNA and survived cells repopulate again. Pancreatic cancer cells, themselves under hypoxic conditions, secrete exosomes that are enriched in miR-301a, which causes M2 polarization of macrophages in the tumor microenvironment. The polarized macrophages enhance tumor growth via the secretion of exosomal miR-501-3p and consequent TGF-β signaling activation. Such polarization fuels progression of pancreatic cancer. Abbreviations: CAF, cancer associated fibroblast; EMT, epithelial mesenchymal transition; NK cell, natural killer cell; PSC, pancreatic stellate cell.

**Table 1 cancers-13-02777-t001:** Role of exosomal miRNAs in pancreatic cancer.

Body Fluids (Source)	Exosomal miRNA	Biomarker or Other Use	References
Serum	miR-17-5p	Diagnostic	[133]
Serum	miR-1246, miR-4644, miR-3976 and miR-4306	Diagnostic	[140]
Serum	miR-10b, miR-21, miR-30c, and miR-181a (up) and miR-let7a (down)	Diagnostic	[141]
Serum	miR-191, miR-21 and miR-451a	Diagnostic	[142]
Serum	miR-21	Early diagnostic; prognostic; recurrence	[133]
Plasma	miR-1246 and miR-196a	Diagnostic	[130,131]
Plasma	miR-483-3p	Diagnostic (differentiation of pancreatic cancer with IPMN)	[143]
Plasma	miR-451a	Prognostic and recurrence	[144]
Blood	miRNA-16a and miRNA-196a	Diagnostic	[145,146]
Pancreatic juice	miR-21 and miR-155	Diagnostic or prognostic	[147]
Saliva	miR-1246 and miR-4644	Diagnostic	[148]
Portal vein blood	miR-4525, miR-451a and miR-21	Prognostic and recurrence	[149]
PCC	miR-23b-3p	Prognostic	[150]
PCC	miR-339-5p	Prognostic	[151]
PCC	miR-222	Prognostic	[52]
PCC	miR-222	Prognostic	[52]
PCC	miR-155 (in vitro and in vivo study)	Prognostic (gemcitabine resistance)	[101]
PCC	miR-27a	Prognostic (promote angiogenesis of human microvascular endothelial cells)	[152]
PCC	miR-212-3p	Prognostic immune suppression (MHC II downregulation)	[153,154]
PCC (under hypoxia)	miR-301a	Prognostic (PTEN/PI3K signaling)	[7]
Dying PCC	miR-194-5p	Prognostic	[114]
M2 macrophage	miR-501-3p	Prognostic (TGF-beta signaling)	[155]
NK cells	miR-3607-3p	Prognostic	[134]
PSC	miR-210	Prognostic (gemcitabine resistance)	[156]
BM-MSC	miR-126-3p	Prognostic (suppress development via ADAM9 down regulation)	[157]
BM-MSC	miR-1231	Prognostic (suppress development)	[158]
UC-MSC	miR-145-5p	Prognostic (suppress progression)	[159]
CAF	miR-146a	Prognostic (gemcitabine resistance)	[160]
CAF	miR-106b	Prognostic (gemcitabine resistance)	[161]

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
