# Peer review of "Exosomal microRNA in Pancreatic Cancer Diagnosis, Prognosis, and Treatment: From Bench to Bedside"

_cancers, 2021, doi:10.3390/cancers13112777_

Round 1

Reviewer 1 Report

I enjoyed reading this well-written review. To further increase scientific merits of this review, I would like to suggest that the authors include following points. PDAC-derived exosomes might also be involved in the development of DM (Javeed et al. Clin. Cancer Res., 21, 1722–1733, 2015; Wang et al. Sci. Rep., 7, 5384, 2017), and Korc (2015) proposed that PDAC-induced DM might represent exosomopathy (Clin. Cancer Res., 21, 1508–1510). Exosomes derived from pancreatic cancer cells induce activation and profibrogenic activities in pancreatic stellate cells (Masamune et al. Biochem Biophys Res Commun . 2018 Jan 1;495(1):71-77). PCC-derived exosomes induce lipolysis in subcutaneous adipose tissue, which might be a novel mechanism of paraneoplastic body weight loss (Sagar et al. Gut 65, 1165-1174, 2016).

Reviewer 2 Report

The review manuscript entitled: Exosomal microRNA in pancreatic cancer diagnosis, prognosis,  and treatment: from bench to bedside, by  Uddin et al., is well written and describes the different features of miRNA as diagnostic,  prognostic or predictive biomarkers when they are encapsulated in exosomes.  However, I found some points to incorporate to the manuscript to improve the quality and soundness of the article:

- In the Introduction section, authors mention "KRAS mutations occurs in more than 90% of cases"; however, there is a controversy in the percentage of KRAS mutations. Please supplement this section with this controversy and provide the rational for which KRAS mutation is one of the cornerstones in PDAC.

- Authors also talk about surgical resection for stage I/II PDAC, please include some statements (and citations)related to the presence of tumor cells and/or KRAS mutation found in peripancreatic fat, which could explain the early progression.

-Please also mention that around 10% of population cannot synthesize CA19-9.

-Since new treatment strategies have appeared based on different drugs combinations (FOLFIRINOX or Gem+Abraxane or Onivyde) or the neoadjuvancy, some benefit has been obtained in survival. Please include information of this combination and the clinical benefit of neoadjuvancy. 

-Then, the following statement: " treatment options for most PDAC patients with metastasis are palliative chemotherapy  and symptomatic treatment " can make a reader feel helpless, and hopeless. Please, re-phase this statement.

-The latin expression et al., must be written in italics.

-Please re-phase the following statement: "Exosomes released by cancer cells can travel to distant organs such as the liver, brain and can establish a metastatic niche to promote proliferation and metastasis", since not only travel to liver and brain and not only set a niche but also prepares microenvironment to allocate metastases.

-Please include a further description of MIF+ exosomes.

-In section " 4.1. Exosomal miRNA in blood samples " authors describe the sensitivity and specificity of miR-10b, miR-21, miR-30c, miR-181a and miR-let7a in blood samples is higher than glypican-1 or CA19-9 levels. Please include further information about this sensitivity and specificity.

-"Exosomal miRNAs also has been isolated  from saliva to find novel biomarkers in gastrointestinal cancer " needs a citation.

-The review provides for comprehensive information on about microRNA. However, some others explanatory and visual figures and tables could improve the manuscript and make it more eye-catching. For example, one figure for each of the prognostic, diagnostic and predictive purposes and a table with Clinical Trials that include miRNA determination.

Round 2

Reviewer 2 Report

Dear authors, the article has been amended accordingly. However, I miss
in the discussion the article by J Kim et al. Gut. 2006
Nov;55(11):1598-605., in which it is described how KRAS mutation is
found in retroperitoneal fat after an R0 resection. Therefore it suggests
that KRAS mutation is delivered to near structures and one of the vias
could be exosomes
